# High Genetic Diversity of HIV-1 and Active Transmission Clusters among Male-to-Male Sexual Contacts (MMSCs) in Zhuhai, China

**DOI:** 10.3390/v15091947

**Published:** 2023-09-18

**Authors:** Yi Zhou, Mingting Cui, Zhongsi Hong, Shaoli Huang, Shuntai Zhou, Hang Lyu, Jiarun Li, Yixiong Lin, Huitao Huang, Weiming Tang, Caijun Sun, Wenyan Huang

**Affiliations:** 1Faculty of Medicine, Macau University of Science and Technology, Macau SAR, China; zhouyi_888@163.com; 2Department of HIV Prevention, Zhuhai Center for Disease Control and Prevention, Zhuhai 519060, China; 930224lh@163.com (H.L.); hht13631256917@163.com (H.H.); 3School of Public Health (Shenzhen), Shenzhen Campus of Sun Yat-sen University, Shenzhen 518107, China; cuimt@mail2.sysu.edu.cn; 4Department of Infectious Diseases, Fifth Affiliated Hospital, Sun Yat-sen University, Zhuhai 519001, China; 5School of Engineering, The Hong Kong University of Science and Technology, Hong Kong 999077, China; 6Lineberger Comprehensive Cancer Center, University of North Carolina at Chapel Hill, Chapel Hill, NC 27599, USA; 7Dermatology Hospital of Southern Medical University, Guangzhou 510315, China; 8Southern Medical University Institute for Global Health and Sexually Transmitted Diseases, Guangzhou 510315, China; 9University of North Carolina Project-China, Guangzhou 510315, China; 10Key Laboratory of Tropical Disease Control (Sun Yat-sen University), Ministry of Education, Guangzhou 510080, China

**Keywords:** human immunodeficiency virus, phylogenetic analysis, transmission network, recent HIV infection, recent infection testing algorithms

## Abstract

Monitoring genetic diversity and recent HIV infections (RHIs) is critical for understanding HIV epidemiology. Here, we report HIV-1 genetic diversity and RHIs in blood samples from 190 HIV-positive MMSCs in Zhuhai, China. MMSCs with newly reported HIV were enrolled from January 2020 to June 2022. A nested PCR was performed to amplify the HIV polymerase gene fragments at HXB2 positions 2604–3606. We constructed genetic transmission network at both 0.5% and 1.5% distance thresholds using the Tamura-Nei93 model. RHIs were identified using a recent infection testing algorithm (RITA) combining limiting antigen avidity enzyme immunoassay (LAg-EIA) assay with clinical data. The results revealed that 19.5% (37/190) were RHIs and 48.4% (92/190) were CRF07_BC. Two clusters were identified at a 0.5% distance threshold. Among them, one was infected with CRF07_BC for the long term, and the other was infected with CRF55_01B recently. We identified a total of 15 clusters at a 1.5% distance threshold. Among them, nine were infected with CRF07_BC subtype, and RHIs were found in 38.8% (19/49) distributed in eight genetic clusters. We identified a large active transmission cluster (*n* = 10) infected with a genetic variant, CRF79_0107. The multivariable logistic regression model showed that clusters were more likely to be RHIs (adjusted OR: 3.64, 95% CI: 1.51~9.01). The RHI algorithm can help to identify recent or ongoing transmission clusters where the prevention tools are mostly needed. Prompt public health measures are needed to contain the further spread of active transmission clusters.

## 1. Introduction

HIV/AIDS remains an important issue in the world with 1.3 million estimated new infections in 2022 [1]. Key populations such as male-to-male sexual contact (MMSC) is disproportionately affected by the HIV epidemic and have a higher risk of infection because of a higher level of risk behavior and/or their vulnerable position in society [2]. In China, the estimated overall national prevalence of HIV among MMSCs from 2001 to 2018 was 5.7% (95% CI: 5.4–6.1%). There has been an observed increasing trend in HIV prevalence over time among MMSCs [3].

Detecting and addressing HIV-infected MMSCs with epidemiological connections is a crucial step toward reducing new HIV infections among MMSCs. HIV transmission clusters, particularly those with recently infected individuals, represent recent or ongoing HIV transmissions in a population. Prompt implementation of prevention strategies targeting active clusters can curb further transmission [4]. HIV-1 undergoes significant diversification and continuous molecular evolution during its global spread [5,6,7]. In recent years, novel circulating recombinant forms (CRFs) and unique recombinant forms (URFs) have been identified in China. It is reported that there are at least 21 CRFs prevalent in China since 2013 [8]. These forms consist of gene segments derived from different HIV-1 genotypes and are predominantly found in regions with a high prevalence of infection and genetic diversity [9,10]. This emergence poses additional challenges to the prevention and control of HIV epidemics. Recombination is rampant within chronic infections and in viral rebound upon antiretroviral therapy (ART) interruption and is an effective way to enable rapid escape from immune selection pressure [11]. Mutations in the specific target of drug action can significantly diminish the efficiency of the drug and weaken the effectiveness of ART in blocking HIV transmission. Zhuhai, located in southern China, is a metropolitan region with high HIV-1 diversity. A previous survey showed that the main HIV-1 genotypes were CRF07_BC (43.3%), CRF01_AE (36.1%) and CRF55_01B (13.4%) among MMSCs in Zhuhai, China [12].

The identification of early HIV infection cases enables us to promptly target active transmission clusters, prioritizing the implementation of prevention tools where they are most needed. In the past years, several laboratory-based assays have been tested to identify early HIV infection according to the natural serological responses after infection [13]. Recently, the World Health Organization (WHO) and the Joint United Nations Programme on HIV/AIDS (UNAIDS) recommend using recent infection testing algorithms (RITAs) to improve the accuracy of identifying recent HIV infection, which integrates HIV recency tests with multiple routinely used clinical assays [14]. In other words, a RITA is a combination of laboratory tests used to classify an HIV infection as recent or long-term. The limiting antigen avidity enzyme immunoassay (LAg-EIA) is one of the widely used serological assays to identify early HIV infection (EHI) [15,16,17]. CD4+ T cell count and the viral load (VL) test are the other two clinical assays used for the majority of RITAs to reclassify recent infection [14]. In comparison to the use of serological assays for the classification of recent HIV infection, RITAs have been proven to accurately classify recent infection cases and effectively reduce the false recent rate (FRR) [14].

Therefore, the major objective of this study was to conduct a comprehensive molecular surveillance of HIV-1 genotypes and recombinants, as well as the identification of the latest RHIs status among MMSCs in Zhuhai. This study aims to provide valuable scientific insights into the development of effective HIV-1 transmission control and prevention strategies in southern China.

## 2. Materials and Methods

### 2.1. Study Design and Population

This implementation study was conducted in Zhuhai, southern China, with an estimated 17,000 MMSCs living in the city, with an HIV prevalence of 7% among MMSCs [18]. There were approximately 150 newly reported HIV-positive cases in the MMSC population in Zhuhai in 2018 [19]. MMSCs with newly reported HIV were enrolled in Zhuhai from January 2020 to June 2022. The inclusion criteria of the study were (1) age ≥ 18 years and (2) new HIV diagnosis without any antiviral treatment. Demographic characteristics (age, ethnicity, marital status, education, occupation, etc.), CD4+ T cell count and the viral loads of participants were extracted from the electronic follow-up records at Zhuhai Center for Disease Control and Prevention (Zhuhai CDC). The study was approved by the institutional reviewing board (IRB) at Zhuhai CDC (ethics approval number No. 2022.11).

### 2.2. PCR Amplification and Recency Testing

HIV diagnostic plasma samples were collected by physicians at diagnostic clinics and transported to Zhuhai CDC for further testing. Viral RNA was extracted from 140 μL plasma using the Roche High Pure Viral RNA Kit (Roche). The kit supplied by Shanghai Huirui Biotechnology Co., Ltd., Shanghai, China. A nested polymerase chain reaction (PCR) was performed to amplify the HIV polymerase (*pol*) gene fragments at HXB2 positions 2604–3606, covering the 528 amino acids of protease and the first 528 amino acids of reverse transcriptase codons. The primers are shown in Table 1. The *pol* sequences were sequenced by Sanger sequencing. We used a RITA combining a limiting antigen avidity enzyme immunoassay (LAg-EIA, supplied by Beijing Kinghawk Pharmaceutical Co., Ltd., Beijing, China) recency assay with clinical data (CD4+ T cell count ≥200 cells/μL and viral load ≥1000 copies/mL) to identify recent HIV infections. The LAg-EIA recency assay includes a preliminary screening test and a confirmation test. Samples with ODn less than or equal to 2.0 in the preliminary screening test were confirmed, and those with an ODn value less than 1.5 in the confirmation test were considered recent infections by LAg-EIA. The mean duration of recent infection for Lag-EIA is 130 days and FRR is 2.3% among MMSCs in China [20]. The FRR of LAg-EIA combined with viral load ≥1000 copies/mL is 4% in Kenya [21].

### 2.3. HIV Sequencing and Subtyping

Sequence fragments were edited and assembled using SeqMan 7.1.0 (44.1). Sequence quality control was performed using HIV-1 sequence quality analysis from the Los Alamos National Laboratory (LANL) (https://www.hiv.lanl.gov/content/sequence/QC/index.html (accessed on 4 December 2022)) and sequences with a proportion of mixed bases greater than 5% were removed. Sequences were aligned and edited by MAFFT (https://www.hiv.lanl.gov/content/sequence/VIRALIGN/viralign.html (accessed on 28 December 2022)), and the length of sequence was 1003 bp (HXB2 position 2604–3606) [22]. Subtyping was verified using phylogenetic methods with reference sequences with a known subtype from the 2020 version of the HIV sequence alignments (LANL) (https://www.hiv.lanl.gov/content/sequence/NEWALIGN/align.html (accessed on 28 December 2022)). The method of maximum likelihood (ML) with GTR + F + I + G4 was performed with the support of the IQTREE 1.6.12 web server [23,24,25]. Clusters with a bootstrap value higher than 0.90 (90%) were defined as the same subtype and subclusters. When genotype results were ambiguous, an RIP online subtyping tool (https://www.hiv.lanl.gov/content/sequence/RIP/RIP.html (accessed on 31 December 2022)) was used to further identify genetic subtypes [26]. The iTol 6.5.2 web server was used for visualization [27].

### 2.4. Phylogenetic and Clustering Analysis

We constructed a genetic transmission network at both 0.5% and 1.5% distance thresholds using the Tamura-Nei93 (TN93) model in HyPhy 2.2.4 using the *pol* region sequences [28,29]. Each patient in the molecular network was represented by a node, and nodes were linked to each other if their pairwise genetic distance was within 0.5% and 1.5% substitutions per site. Network visualization was undertaken by using Cytoscape 3.7.0 [30]. We performed analysis for different distance thresholds (ranging from 0.1% to 3.25%) to determine the optimal threshold (i.e., the distance when the maximum ratio of the number of clusters to distance is detected) using MicrobeTrace (https://github.com/CDCgov/MicrobeTrace/wiki (accessed on 26 August 2023)) [31]. We ranked the edgecounts of all nodes within the network, and defined nodes with edgecounts greater than or equal to the upper quartile (i.e., edgecount ≥ 5) as having high linkage. Nodes with high linkage were defined as individuals with a high risk of transmission. CRF79_0107 sequences and their information were downloaded from LANL (https://www.hiv.lanl.gov/components/sequence/HIV/geo/geo.html (accessed on 26 June 2023)) [32].

To investigate interprovincial and intraprovincial clusters, we downloaded sequences with the highest homology to those in this study using HIV BLAST (https://www.hiv.lanl.gov/content/sequence/BASIC_BLAST/basic_blast.html (accessed on 26 August 2023)) [33], the top 10 sequences for those identified as CRF79_0107 in this study and the top 5 sequences for other variants. Location data were gathered, and sequences with 99% or higher HIV BLAST identity were included. The coordinates of the provincial capital city would be utilized only if the province’s geographic location was provided. These sequences and sequences of this study were aligned using MAFFT and edited by Aliview [34]. Then, we constructed a genetic transmission network at a 1.5% distance threshold using the TN93 model (R package “ape”) and MicrobeTrace.

### 2.5. Statistical Analysis

Descriptive analysis was performed to describe the socio-demographics and gene subtypes of participants using R 4.1.2. R packages, including “mapdata”, “maps”, “maptools”, “rgdal”, “ggplot2”, “plyr”, “cowplot” and “sf”, were used to plot and display maps depicting the distribution of samples. R package “ape” was used to calculate genetic distance using the TN93 model. R packages including “tidyverse” and “dplyr” were used for data manipulation and visualization. R package “broom” was used for multivariable analysis. Categorical variables were described by frequency (*n*) and percentage (%). Statistical comparisons of the recent infections and long-term infections were performed using the Chi-square test or Fisher’s exact test. All tests were two-tailed, and values of *p* < 0.05 were considered statistically significant. Preliminary analyses were conducted to fit unadjusted and adjusted logistic regression models of MMSC characteristic covariates on each of the two outcomes of RITA. For these models, Y was the outcome of RITA and Y = 1 when the outcome was recent infection (versus Y = 0 in the case of long-term infection).

## 3. Results

### 3.1. Demographics and HIV Genotyping of Study Subjects

A total of 190 (71.7%, 190/265) participants were enrolled with successful sequencing of the HIV-1 *pol* gene in Zhuhai from January 2020 to June 2022. All sequences exhibited a proportion of mixed bases below 0.5%. Among the 190 participants, 65.3% (124/190) were between 18 and 35 years old, 91.6% (174/190) were of Han ethnicity, 72.6% (138/190) were single, 42.6% (81/190) had obtained a college degree or higher, and 37.9% (72/190) were office workers. The recency assay revealed that 19.5% (37/190) were recently infected (Table 2). Of the 190 participants, 48.4% (92/190) were CRF07_BC, 17.9% (34/190) were CRF55_01B, 16.3% (31/190) were CRF01_AE, 10% (19/190) were CRF79_0107 and others (B, CRF08_BC, CRF103_01B, CRF104_0107, CRF15_01B, CRF59_01B, CRF68_01B, CRF76_01B, CRF80_0107 and URF) accounted for 7.4% (14/190) (Table 2 and Figure 1). MMSCs with high viral loads were more likely to be clustering insignificantly (Table 3).

### 3.2. Molecular Network Analysis and Active Transmission Clusters

According to previous studies and the analysis of distance thresholds ranging from 0.1% to 3.25%, a threshold of 1.5% was optimal for detecting transmission clusters (Figure 2A) [35,36]. Using the threshold of 0.5% genetic distance, 2 clusters were identified, and 3.7% (7/190) of the participants could be linked to a genetic cluster. Among the 2 clusters, 1 was infected with CRF07_BC, and the other with CRF55_01B. Using the threshold of 1.5% genetic distance, we identified a total of 15 clusters (median size 2, range: 2–10), and 25.8% (49/190) of the participants could be linked to a genetic cluster. Among the 15 clusters, 9 were infected with CRF07_BC subtype, 4 with CFR55_01B and 2 with CRF79_0107. RHIs were found in 38.8% (19/49) of the participants distributed in 8 genetic clusters, i.e., active transmission clusters. The largest cluster we identified (cluster #1) had 10 members, and 60% of them were recently infected. Of note, this cluster was infected with CRF79_0107. The molecular network diagram of Zhuhai is shown in Figure 2.

To further understand the epidemic of CRF79_0107, we downloaded all CRF79_0107 sequences (41 sequences) and their basic information including accession, blood collection year, country, province and city from LANL. All CRF79_0107 sequences were detected in China, and the first CRF79_0107 sequence was identified in 2017 from Shanxi, China. The earliest blood collection year of the CRF79_0107 sequences identified in Shanxi was 2015. There were three male cases identified as CRF79_0107 in Shanxi in 2015, whose risk factors were MMSC (2/3) or heterosexual contact (1/3). In 2021, a female case whose blood was collected in 2011 from Shenzhen, China, was identified as CRF79_0107. The CRF79_0107 sequences were mainly identified in the north of China (Figure 3). In southern China, only Shenzhen and Zhuhai reported CRF79_0107. The distribution of sex and risk factors of CRF79_0107 is shown in Figure 4. Men accounted for 65% (39/60) of CRF79_0107 sequences in the HIV database and this study, and 71.8% (28/39) were MMSCs.

A total of 110 sequences were included in the analysis to investigate interprovincial and intraprovincial clusters. At a 1.5% genetic distance threshold, 15 clusters were identified. Interprovincial clusters revealed connections between sequences from Zhuhai identified as CRF07_BC, CRF55_01B or CRF79_0107 and sequences from other cities in Guangdong province (Figure 5A–D). On the other hand, intraprovincial clusters demonstrated that sequences from Zhuhai were also connected, either indirectly or directly, with other provinces in China (Figure 5E,F). Within the HIV transmission network in Zhuhai, some sequences from Zhuhai were observed as singletons, indicating unique transmission events within the city (Figure 2C). However, these sequences clustered and shared links with sequences from other cities in Guangdong province or other provinces (Figure 5G).

We further explored the factors that were associated with recent HIV infection at diagnosis using a multivariable logistic regression model (Table 4). Participants clustering (compared with not clustering, adjusted OR [aOR]: 3.64, 95% CI: 1.51~9.01) and students (aOR: 6.47, 95% CI: 1.71~25.66) were more likely to be recently infected. Notably, we observed a trend that variant, CRF79_0107, was associated with recent infection (*p* = 0.057). These findings suggests that there are ongoing active HIV transmissions among young students, particularly the variant, CRF79_0107.

We defined nodes with an edgecount ≥5 as those with high linkage. Fisher’s exact test showed that age, ethnicity, marriage and subtype were significant factors associated with high linkage (Table 5). MMSCs with high linkage were more likely to be younger, single and non-Han.

## 4. Discussion

This study holds significant importance as it meticulously outlines the molecular epidemiology of HIV-1 among recently diagnosed MMSCs in Zhuhai, a bustling metropolitan area located in southern China. The observed extensive genetic diversity within HIV-1 strains strongly implies intricate introductions of the virus among MMSCs in the Zhuhai region. Guangzhou, Shenzhen and Zhuhai are three metropolitan areas with a high percentage of migrant population in the Guangdong–Hong Kong–Macao Greater Bay Area, and the most popular working and living destination for MMSCs. The most prevalent HIV-1 variants circulating among MMSCs in Guangzhou was CRF07_BC (41.6%), followed by CRF01_AE (30.0%) and CRF55_01B (12.8%) [37], while in the neighboring city of Shenzhen, the most prevalent variant is CRF07_BC (41.12%), followed by CRF01_AE (35.14%) and CRF55_01B (11.23%) [38]. Our study suggested that CRF07_BC was the most commonly seen HIV-1 variant circulating among MMSCs in Zhuhai, which is consistent with the findings in Guangzhou and Shenzhen, but other subtypes such as CRF55_01B and CRF01_AE differ among these cities.

In a previous study, we observed that CRF79_0107 accounted for only 0.09% (9/10378) of newly diagnosed HIV-1 infections in Shenzhen [38]. However, the outbreak of CRF79_0107 identified in this study has raised concerns regarding the rapid transmission of this specific variant. CRF79_0107 was first identified from Shanxi, China in 2017 [39], and in 2021, a sequence from a blood specimen collected from a female participant in Shenzhen was also identified as CRF79_0107 [38]. Our study identified a total of 19 CRF79_0107 sequences. The emergence of CRF79_0107 increases the complexity of the HIV epidemic. From the analyses of CRF79_0107 data from LANL, we suspect that CRF79_0107 initially spread through heterosexual contact and was later introduced to bisexual males, eventually becoming prevalent among MMSCs. Furthermore, it is evident that CRF79_0107 exhibited a predominant presence in regions characterized by robust economic development and notable seasonal human migration patterns. Consequently, there was a potential that CRF79_0107 might have the tendency to spread to the MMSC population in surrounding areas, akin to the pattern observed with CRF55_01B [40]. In our study, participants identified as CRF79_0107 within the network accounted for 46.2% (6/13) of those with high linkage and 16.7% (6/36) of those with low linkage. Participants with high linkage were defined as individuals with a high risk of transmitting HIV. Therefore, it suggested that CRF79_0107 might be transmitted rapidly in Zhuhai, although the sample size of nodes within the network was limited.

In our study, we identified a total of 15 transmission clusters from 49 individuals and 5 clusters with more than 2 members using the 1.5% threshold. Using a more specific threshold of 0.5%, we only identified 2 clusters. This finding suggests that the majority of the participants diagnosed were not identified within a known transmission cluster. The low cluster rate indicated that early treatment might have played an important role in the epidemic of HIV in the study population because of “U=U” (undetectable equals untransmissible). The initiation of ART in early HIV infection regardless of CD4+ count provided net benefits for individual patients in reducing the risk of viral transmission [41]. From 2014 and 2021, the percentage of HIV-infected persons receiving ART increased from 59.0% to 96% in Zhuhai [42,43,44]. Viral load was higher in people in larger clusters and with increased network connectivity [45], but the proportion of cluster members with unsuppressed viral load showed only a weak association (HR, 1.35 (95% CI, 0.98–1.86)) with incident cluster growth [46]. The data indicated a correlation between elevated viral loads and increased transmission rates among patients. However, upon their identification, the transmission rate tended to decline. This was attributed to the prompt initiation of ART in these individuals, typically resulting in the suppression of viral levels to undetectable measures. Thus, the majority of participants in this study appeared to be singletons in the transmission network. However, the presence of recent infection or active transmission networks remains significant, emphasizing the importance of enhancing early detection and treatment of HIV.

Our study found that recent infections were 38.8% (19/49) of the participants distributed in 8 genetic clusters, i.e., active transmission clusters (of the 1.5% cutoff). Additionally, our investigation unveiled that 13.1% of cases were indicative of recent infections, yet they did not exhibit any discernible clustering. This observation suggests the presence of unidentified transmission clusters in Zhuhai or the possibility that these clusters span multiple cities. RHIs are often linked to active transmission networks. The quick identification and intervention of recently infected individuals is crucial in HIV monitoring and prevention [47,48]. Within an active transmission cluster, the term RHI pertains to an individual who contracted HIV either from a diagnosed patient with uncontrolled viral load or from an individual living with HIV whose status has not been officially confirmed. Immediate treatment for individuals diagnosed with a new HIV diagnosis is available in Zhuhai; thus, it is likely that a diagnosed patient in Zhuhai is on ART. There are two potential reasons why the viral load of the patient under ART might remain unsuppressed: (1) the patient has recently initiated ART, or (2) the patient has treatment failure because of poor adherence or drug resistance mutations [49]. In the first situation, it is important to provide the patients with the information regarding the relationship between viral load and HIV transmission [50]. It is advisable for them to avoid participating in unprotected sexual activities until their viral load is properly suppressed. It should be noted that these observations are based on findings from another study and not our own results. In the second scenario, close monitoring of viral load and drug resistance testing by healthcare professionals are essential to ensure the efficacy of ART and the prevention of potential transmission [51]. Public health measures are needed to identify individuals with undiagnosed HIV, particularly when there is suspicion that they are linked to RHIs. Therefore, active transmission clusters and individuals with recent infections need to be prioritized for prompt public health intervention to contain further HIV transmission. However, it remains challenging to use the cluster detection to prospectively identify and interrupt incident transmission events in epidemics [46]. Cluster formation requires a sufficient number of newly infected individuals in the population, which may take time to reach detectable levels during the early stages of an epidemic with low infection rates. This time delay impedes the prompt identification and response to incident transmission events. Additionally, cluster detection relying solely on genetic sequence data may not provide a comprehensive understanding of transmission dynamics. Augmenting genetic data with additional epidemiological information, such as behavioral characteristics, contact networks and demographic data, is often necessary for accurate identification and tracing of incident transmission events. The absence of such data can hinder the effectiveness of cluster detection approaches.

On the other hand, the number of HIV-infected people in the interprovincial transmission cluster exceeds that in the intraprovincial cluster, suggesting that the migration of people among provinces and cities plays an important role in shaping the prevalence of the AIDS epidemic in China [52]. Therefore, it is crucial to address the mobility of the MMSC population and adopt strategies such as digital-based HIV self-testing to engage with molecular clusters or risk networks to prevent new infections [18].

The key indicator of the success of the targeted prevention approaches is the HIV incidence, which can be estimated by HIV recency tests. WHO and UNAIDS recommend using RITAs, which integrate HIV recency tests with multiple routinely used clinical assays, to improve the accuracy of identifying recent HIV infection [14]. We applied the RITA approach to identify RHIs in our study, but it has several limitations. The manufacturer of the LAg-Avidity assay recommends excluding individuals who are receiving ART or elite suppressors or have AIDS (CD4 cell count < 200 cells/μL) from incidence surveys. The results of this study indicated that those exclusions did not remove all sources of assay misclassification among individuals with long-standing HIV infection [53]. RITAs can identify RHIs in epidemiological studies with a population. RHIs are of particular interest in prevention efforts because they are more likely to be actively transmitting the virus [54,55]. In addition to RITAs, genetic sequencing plays a crucial role in hotspot identification. This information helps identify clusters of infections, indicating potential transmission networks and hotspots. By combining RITAs and genetic sequencing, researchers can pinpoint geographic areas or populations with a higher concentration of RHIs or genetically related viral strains [56]. These areas or populations may represent key hotspots for targeted prevention efforts. By focusing resources, interventions and education on these hotspots, public health officials can effectively allocate their efforts to reduce the spread of the infection and implement prevention strategies tailored to the specific needs of the affected communities.

Our study has a few limitations. First, it should be noted that the samples collected by convenience did not represent a random sampling. In addition, we were unable to obtain sequences from some participants. Thus, an unknown degree of sampling bias might exist in our study. Second, we did not identify significant differences between the recent infection group and the long-term infection group at baseline, indicating that the sample size of this study might be small. Lastly, we used only partial genetic sequences instead of complete genomes for the phylogenetic analysis, which might be suboptimal in areas with high HIV genetic variations.

## 5. Conclusions

We identified an active transmission of the HIV-1 variant in the study area. The recent infection testing algorithm can help to identify recent or ongoing transmission clusters where the prevention tools are mostly needed. Prompt public health measures are needed to contain the further spread of active transmission clusters.

## Figures and Tables

**Figure 1 viruses-15-01947-f001:**
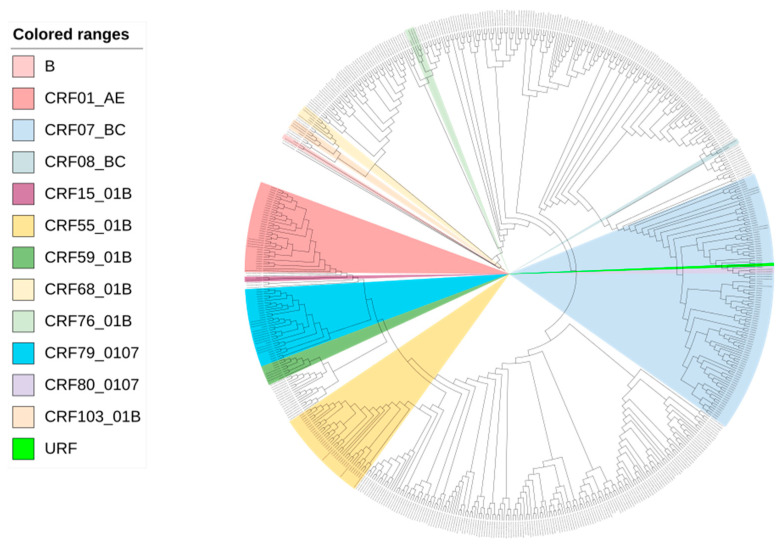
Maximum-likelihood phylogenetic tree of 190 newly reported MMSCs with HIV in Zhuhai from 2020 to 2022.

**Figure 2 viruses-15-01947-f002:**
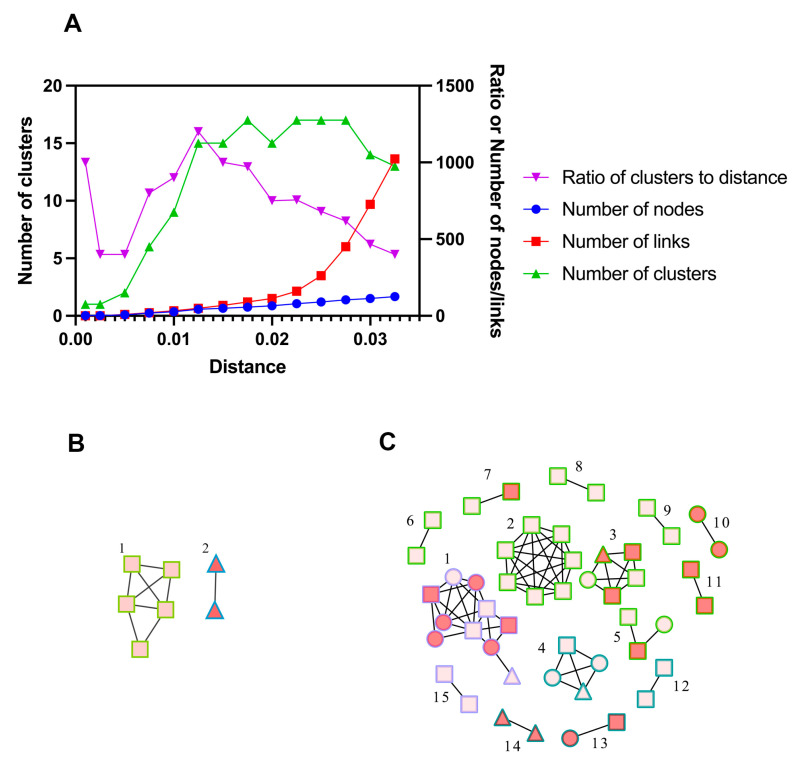
(**A**) Selection of the optimal distance threshold. Curves represented the number of clusters, the number of nodes, the number of links, or the ratio of the number of clusters to distance at different distance thresholds. (**B**,**C**) The molecular transmission cluster diagram of newly reported MMSCs with HIV in Zhuhai from 2020 to 2022. ○: 2020. □: 2021. △: 2022. Red nodes represented recent infections and pink nodes represented long-term infections. Green border represented CRF07_BC, blue border represented CRF55_01B, and purple border represented CRF79_0107. (**B**) Using the threshold of 0.5% genetic distance. (**C**) Using the threshold of 1.5% genetic distance. Cluster #1 at a 0.5% distance threshold was from cluster #2 at a 1.5% distance threshold. Cluster #2 at a 0.5% distance threshold was from cluster #14 at a 1.5% distance threshold.

**Figure 3 viruses-15-01947-f003:**
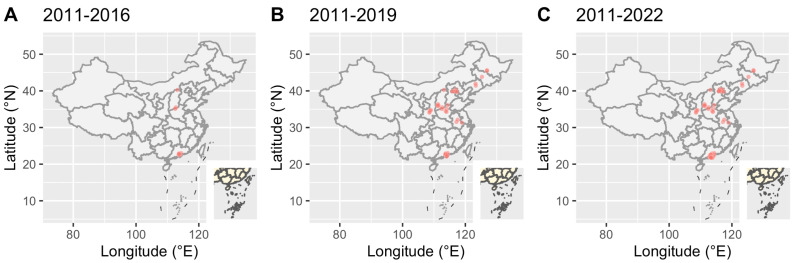
Geographic distribution of CRF79_0107 in the HIV database and this study over time. Red dots represented HIV-infectious cases.

**Figure 4 viruses-15-01947-f004:**
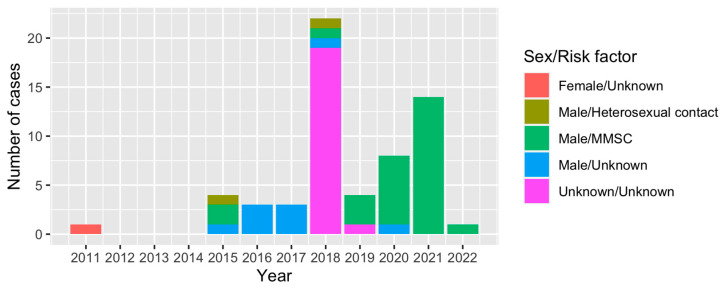
Distribution of sex and risk factors of cases identified as CRF79_0107 in the HIV database and this study over time.

**Figure 5 viruses-15-01947-f005:**
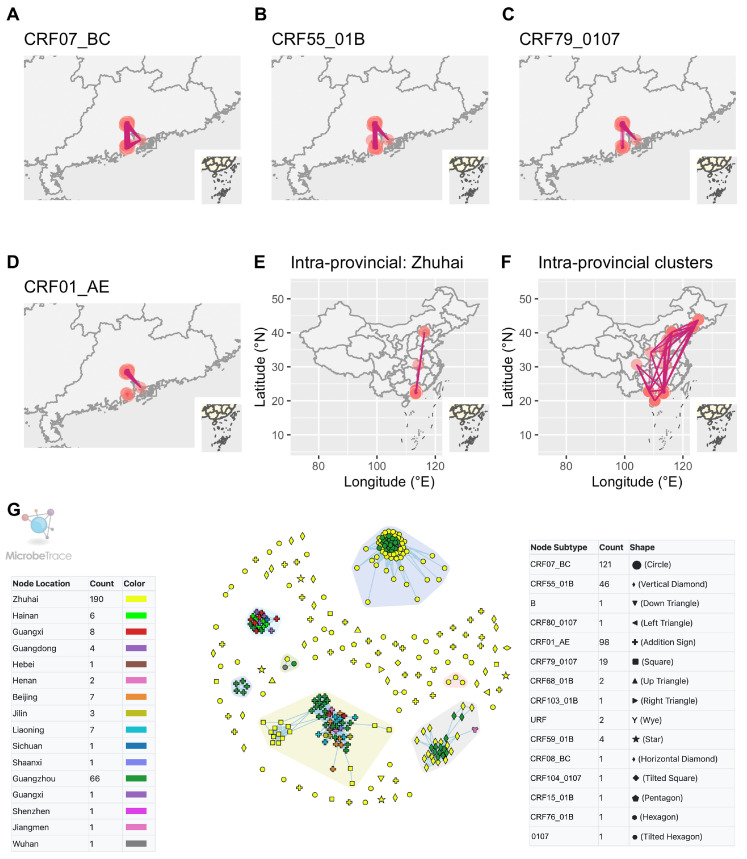
Interprovincial and intraprovincial links between Zhuhai and other provinces in China. (**A**–**D**) Links between sequences of other cities in Guangdong and different CRFs of Zhuhai. (**E**) Links between sequences of other provinces and those of Zhuhai. (**F**) Links among sequences of China from NCBI and those of Zhuhai. (**G**) All included sequences in HIV-1 transmission network. The clusters in the colored polygons indicated the presence of more than two nodes within each cluster. Zhuhai, Guangzhou, Shenzhen and Jiangmen are cities in Guangdong province.

**Table 1 viruses-15-01947-t001:** List of the PCR primers used in the study.

Name	Sequences (5′-3′)	Position in HBX2
HR26871	TTAAAGCCAGGAATGGATGG	2583–2602
HR26872	TCAGGATGGAGTTCATACCCC	3259–3239
HR26873	TGGAAAGGATCACCAGCAATA	3006–3026
HR27372	TGTTAACTGTCTTACATCATTAGTGTG	3704–3681

**Table 2 viruses-15-01947-t002:** Sociodemographic characteristics of 190 HIV-positive MMSCs.

Variable	Level	Long-Term Infection (*n* = 153, 80.5%)	Recent Infection (*n* = 37, 19.5%)	Total (*n* = 190, 100%)	*p*
Age (year)	18~35	96 (62.7)	28 (75.7)	124 (65.3)	0.197
	>35	57 (37.3)	9 (24.3)	66 (34.7)	
Ethnicity	Han	141 (92.2)	33 (89.2)	174 (91.6)	0.520 ^1^
	Others	12 (7.8)	4 (10.8)	16 (8.4)	
Marriage	Single	109 (71.2)	29 (78.4)	138 (72.6)	0.193 ^1^
	Married	25 (16.3)	2 (5.4)	27 (14.2)	
	Separated or divorced	19 (12.4)	6 (16.2)	25 (13.2)	
Education	Junior high school and below	42 (27.5)	7 (18.9)	49 (25.8)	0.493
	High school	46 (30.1)	14 (37.8)	60 (31.6)	
	College degree or above	65 (42.5)	16 (43.2)	81 (42.6)	
Occupation	Office worker	61 (39.9)	11 (29.7)	72 (37.9)	0.195 ^1^
	Housework or unemployment	30 (19.6)	7 (18.9)	37 (19.5)	
	Ordinary worker or farmer	28 (18.3)	6 (16.2)	34 (17.9)	
	Sexual worker	19 (12.4)	4 (10.8)	23 (12.1)	
	Student	8 (5.2)	7 (18.9)	15 (7.9)	
	Others	7 (4.6)	2 (5.4)	9 (4.7)	
Subtype	CRF07_BC	75 (49.0)	17 (45.9)	92 (48.4)	0.041 ^1^
	CRF55_01B	26 (17.0)	8 (21.6)	34 (17.9)	
	CRF01_AE	29 (19.0)	2 (5.4)	31 (16.3)	
	CRF79_0107	11 (7.2)	8 (21.6)	19 (10.0)	
	Others	12 (7.8)	2 (5.4)	14 (7.4)	
Cluster ^2^	No	123 (80.4)	18 (48.6)	141 (74.2)	<0.001
	Yes	30 (19.6)	19 (51.4)	49 (25.8)	

^1^ *p* was calculated with Fisher’s exact test. The other *p*-values were calculated with Chi-square test. ^2^ Clusters were identified at a 1.5% distance threshold.

**Table 3 viruses-15-01947-t003:** The association between viral load and clustering of 190 HIV-positive MMSCs.

Viral Load (copies/mL)	Nonclustering ^2^ (*n* = 141, 74.2%)	Clustering ^2^ (*n* = 49, 25.8%)	Total (*n* = 190, 100%)	*p* ^1^
<1000	52 (36.9)	10 (20.4)	62 (32.6)	0.091
1000~10,000	10 (7.1)	3 (6.1)	13 (6.8)	
≥10,000	72 (51.1)	35 (71.4)	107 (56.3)	
Missing	7 (5.0)	1 (2.0)	8 (4.2)	

^1^ *p* was calculated with Fisher’s exact test. ^2^ Clusters were identified at a 1.5% distance threshold.

**Table 4 viruses-15-01947-t004:** Univariable analysis and multivariable analysis for recent infection among HIV-positive MMSCs: logistic regression analysis.

Variable	Level	Univariable Analysis		Multivariable Analysis	
		OR (95%CI)	*p*	aOR (95%CI)	*p*
Age (year)	18~35	ref.			
	>35	0.54 [0.23, 1.19]	0.142		
Ethnicity	Han				
	Others	1.42 [0.38, 4.39]	0.561		
Marriage	Single	ref.			
	Married	0.30 [0.05, 1.09]	0.116		
	Separated or divorced	1.19 [0.40, 3.10]	0.738		
Education	Junior high school and below	ref.			
	High school	1.83 [0.69, 5.22]	0.237		
	College degree or above	1.48 [0.58, 4.12]	0.430		
Occupation	Office worker	ref.		ref.	
	Housework or unemployment	1.29 [0.44, 3.63]	0.628	1.71 [0.53, 5.41]	0.359
	Worker or farmer	1.19 [0.38, 3.46]	0.757	1.96 [0.57, 6.51]	0.270
	Sexual worker	1.17 [0.30, 3.87]	0.809	1.71 [0.40, 6.60]	0.442
	Student	4.85 [1.44, 16.45]	0.010	6.47 [1.71, 25.66]	0.006
	Others	1.58 [0.22, 7.66]	0.595	4.03 [0.51, 22.87]	0.134
Subtype	CRF07_BC	ref.		ref.	
	CRF55_01B	1.36 [0.50, 3.44]	0.529	1.38 [0.46, 3.91]	0.546
	CRF01_AE	0.30 [0.05, 1.15]	0.127	0.47 [0.07, 1.97]	0.355
	CRF79_0107	3.21 [1.10, 9.20]	0.030	3.03 [0.95, 9.56]	0.057
	Others	0.74 [0.11, 3.04]	0.704	1.29 [0.18, 5.95]	0.764
Cluster *	No	ref.		ref.	
	Yes	4.33 [2.03, 9.33]	<0.001	3.64 [1.51, 9.01]	0.004

* Clusters were identified at a 1.5% distance threshold.

**Table 5 viruses-15-01947-t005:** Factors associated with high linkage among clustered individuals within networks.

Variable	Level	MMSCs with High Linkage (*n* = 13, 26.5%)	MMSCs with Low Linkage (*n* = 36, 73.4%)	Total (*n* = 49, 100%)	*p **
Age (year)	18~35	13 (100.0)	25 (69.4)	38 (77.6)	0.024
	>35	0 (0.0)	11 (30.6)	11 (22.4)	
Ethnicity	Han	9 (69.2)	34 (94.4)	43 (87.8)	0.036
	Others	4 (30.8)	2 (5.6)	6 (12.2)	
Marriage	Single	13 (100.0)	23 (63.9)	36 (73.5)	0.033
	Married	0 (0.0)	4 (11.1)	4 (8.2)	
	Separated or divorced	0 (0.0)	9 (25.0)	9 (18.4)	
Education	Junior high school and below	1 (7.7)	8 (22.2)	9 (18.4)	0.300
	High school	2 (15.4)	10 (27.8)	12 (24.5)	
	College degree or above	10 (76.9)	18 (50.0)	28 (57.1)	
Occupation	Office worker	9 (69.2)	14 (38.9)	23 (46.9)	0.246
	Housework or unemployment	2 (15.4)	7 (19.4)	9 (18.4)	
	Ordinary worker or farmer	0 (0.0)	5 (13.9)	5 (10.2)	
	Sexual worker	0 (0.0)	6 (16.7)	6 (12.2)	
	Student	2 (15.4)	4 (11.1)	6 (12.2)	
Subtype	CRF07_BC	7 (53.8)	20 (55.6)	27 (55.1)	0.031
	CRF55_01B	0 (0.0)	10 (27.8)	10 (20.4)	
	CRF79_0107	6 (46.2)	6 (16.7)	12 (24.5)	
Recency	Long-term infection	10 (76.9)	20 (55.6)	30 (61.2)	0.205
	Recent infection	3 (23.1)	16 (44.4)	19 (38.8)	

*p* *-value was calculated with Fisher’s exact test.

## Data Availability

The raw data are not publicly available to protect individuals’ privacy and because of restrictions imposed by the data use agreements.

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
