# Peer review of "High Genetic Diversity of HIV-1 and Active Transmission Clusters among Male-to-Male Sexual Contacts (MMSCs) in Zhuhai, China"

_viruses, 2023, doi:10.3390/v15091947_

Round 1

Reviewer 1 Report

The reviewer has no significant remarks on the manuscript. The study was carried out in a classical style. Reasonably and logically used standard laboratory and bioinformatics methods of analysis. The results obtained are described qualitatively and clearly, the conclusions drawn are justified.

However, there are a few minor remarks that require corrections.

1. The authors in several places (lines 36 and 158) mistakenly write CRF01_BC instead of CFR07_BC. It is necessary to correct this, and also check the other names of CRFs im manuscript.

2. In line 37, the authors refer to CRF79_0107 as a new subtype. This is an unfortunate formulation, since for HIV there are exact definitions of what is a subtype. In this case, it is better to use the wording "genetic variant" for example.

3. Authors indicate that as part of their work, they performed analysis and received 190 nucleotide sequences of HIV. In addition, the authors write that one of the goals of the work was to develop effective monitoring of HIV transmission and prevention strategies. At the same time, the authors did not deposit the resulting HIV sequences in public databases (Genbank). This reduces the significance of the work, since others will not be able to apply this information in their future work in the direction of the HIV molecular epidemiology.

Author Response

  1. The authors in several places (lines 36 and 158) mistakenly write CRF01_BC instead of CFR07_BC. It is necessary to correct this, and also check the other names of CRFs im manuscript.

We thank the reviewer for the suggestion. These mistakes were revised in the resubmitted manuscript.

  1. In line 37, the authors refer to CRF79_0107 as a new subtype. This is an unfortunate formulation, since for HIV there are exact definitions of what is a subtype. In this case, it is better to use the wording "genetic variant" for example.

We thank the reviewer for the suggestion. This mistake was revised in the resubmitted manuscript.

  1. Authors indicate that as part of their work, they performed analysis and received 190 nucleotide sequences of HIV. In addition, the authors write that one of the goals of the work was to develop effective monitoring of HIV transmission and prevention strategies. At the same time, the authors did not deposit the resulting HIV sequences in public databases (Genbank). This reduces the significance of the work, since others will not be able to apply this information in their future work in the direction of the HIV molecular epidemiology.

We thank the reviewer for the suggestion. The raw data are not publicly available to protect individuals’ privacy and due to restrictions imposed by the data use agreements.

Reviewer 2 Report

Zhou et al. conducted molecular surveillance of HIV-1 genotypes and recombinants, as well as tested for recent infection status, among MSM in Zhuhai. A number of findings are important as part of this study, including a large fraction (>10%) of recent infections that did not contribute to transmission clusters, suggesting ongoing transmission clusters are under-investigated. Additionally, the association of young students with these recent infections is important. However, I do have some major concerns that need to be addressed.

Major comments:

Overall:

·       Grammar issues throughout this manuscript (particularly in the discussion) make it very difficult to read.

Abstract:

·       Need to define RITA (line 33)

·       “Prompt public health measures are needed to contain the outbreak of CRF79_0107 in China.” (lines 40-41)

I don’t think there is enough evidence to support this, considering the p-value for the multiple logistic regression >0.05 and only 21% of RHIs were typed as CRF79_0107 (Table 2). Wording throughout seems to suggest that this statement at least in part based on the evidence that one (“large”) cluster comprised this subtype; however, large clusters are not necessarily a sign of faster spread – sampling bias can play a big role here.

Introduction:

·        “HIV transmission clusters represent recent or ongoing HIV transmissions in a population” (line 55) I would be careful here. If you’re referring to clusters identified using genetic clustering techniques, the results reflect groups of individuals with smaller pairwise genetic distances, which can be explained by sampling of direct transmission chains/networks or also more sparse sampling of individuals exhibiting rapid transmission. In fact, older clusters that are not ongoing (at least in the samples collected) will still show up using these techniques. In fact, in your own study, you find that only 51% of clustered individuals were RHIs and almost half of the clusters did not contain any RHIs. A rephrasing here is all that is needed but is necessary.

Materials and Methods:

·       What is the false positive rate of this RITA? This should also be discussed in light of the results.

Results:

·       What does the p-value in Table 2 actually represent?

Discussion:

·       “In the previous study, we found that CRF79_0107 accounted for only 0.09%  (9/10378) HIV-1 newly diagnosed infections in Shenzhen [27]. The outbreak of CRF79_0107 identified in this study raised the concern of the rapid transmission of this specific subtype.” (lines 210-212). Perhaps more analyses could be done to help rule out sampling bias – i.e., did demographics differ between studies?

·       What is meant by “lower network rate” in line 223?

·       “The number of transmission clusters also infers that antiretroviral therapy was an effective prevention approach to prevent HIV transmission among  MSM in Zhuhai.” (Lines 229-230)

Is this compared to previous transmission cluster numbers or prevalence? If so, please cite and present these data.

·       “We did not identify significant differences between the recent infection group and long-term infection group at baseline, indicating a small sample size in this study.” (Lines 266-267). What differences were evaluated? This was not mentioned in the results or methods.

Minor comments:

Materials and Methods:

·       A justification of the 1.5% cutoff should be included, not just a reference to earlier uses.

Discussion:

·       HARRT should be HAART (line 225), and its true reference stated (highly active antiretroviral therapy. Otherwise, just use ART abbreviation.

As stated above, grammar issues throughout this manuscript (particularly in the discussion) make it very difficult to read.

Author Response

Major comments:

Overall:

  • Grammar issues throughout this manuscript (particularly in the discussion) make it very difficult to read.

Abstract:

  • Need to define RITA (line 33)

 RITA was defined in the resubmitted manuscript.

  • “Prompt public health measures are needed to contain the outbreak of CRF79_0107 in China.” (lines 40-41)

I don’t think there is enough evidence to support this, considering the p-value for the multiple logistic regression >0.05 and only 21% of RHIs were typed as CRF79_0107 (Table 2). Wording throughout seems to suggest that this statement at least in part based on the evidence that one (“large”) cluster comprised this subtype; however, large clusters are not necessarily a sign of faster spread – sampling bias can play a big role here.

 We thank the reviewer for the suggestion. Prompt public health measures are needed to contain the further spread of active transmission clusters.

Introduction:

  • “HIV transmission clusters represent recent or ongoing HIV transmissions in a population” (line 55) I would be careful here. If you’re referring to clusters identified using genetic clustering techniques, the results reflect groups of individuals with smaller pairwise genetic distances, which can be explained by sampling of direct transmission chains/networks or also more sparse sampling of individuals exhibiting rapid transmission. In fact, older clusters that are not ongoing (at least in the samples collected) will still show up using these techniques. In fact, in your own study, you find that only 51% of clustered individuals were RHIs and almost half of the clusters did not contain any RHIs. A rephrasing here is all that is needed but is necessary.

We are sorry for our confusing description. HIV transmission clusters which newly reported recent infectious cases were added to over time represent recent or ongoing HIV transmissions in a population, where prompt implementation of prevention strategies can curb further transmission.

Materials and Methods:

  • What is the false positive rate of this RITA? This should also be discussed in light of the results.

 False recent rate of LAg-EIA combined with viral load ≥1000 copies/mL is 4%. There are no false recent rate of LAg-EIA combined with viral load ≥1000 copies/mL and CD4+ T cell count ≥200 cells/μL estimated. But WHO recommended recency judgement combined with viral load and CD4+ T cell count and many studies used criteria including LAg-EIA, viral load ≥1000 copies/mL and CD4+ T cell count ≥200 cells/μL. Because in this study, participants could not be defined as true RHI or long-term infection (based on repeat testing and follow-up), we did not discuss false recent rate in the manuscript.

Results:

  • What does the p-value in Table 2 actually represent?

If p-value < 0.05, it means HIV infection recency is more likely to be associated with this variable.

Discussion:

  • “In the previous study, we found that CRF79_0107 accounted for only 0.09%  (9/10378) HIV-1 newly diagnosed infections in Shenzhen [27]. The outbreak of CRF79_0107 identified in this study raised the concern of the rapid transmission of this specific subtype.” (lines 210-212). Perhaps more analyses could be done to help rule out sampling bias – i.e., did demographics differ between studies?

We analyzed the association between CRF79_0107 and a high risk of transmitting HIV.

  • What is meant by “lower network rate” in line 223?

 We are sorry for confusing description. “Lower network rate” was replaced by “lower cluster rate” which was the ratio of the number of participants connected within networks to the total number of participants.

  • “The number of transmission clusters also infers that antiretroviral therapy was an effective prevention approach to prevent HIV transmission among  MSM in Zhuhai.” (Lines 229-230)

Is this compared to previous transmission cluster numbers or prevalence? If so, please cite and present these data.

 Viral load was higher in people in larger clusters and with increased network connectivity [WERTHEIM J O, OSTER A M, SWITZER W M, et al. Natural selection favoring more transmissible HIV detected in United States molecular transmission network [J]. Nat Commun, 2019, 10(1): 5788.10.1038/s41467-019-13723-z]. But the proportion of cluster members with unsuppressed viral load showed only a weak association (HR, 1.35 [95% CI, 0.98–1.86]) with incident cluster growth [LITTLE S J, CHEN T, WANG R, et al. Effective Human Immunodeficiency Virus Molecular Surveillance Requires Identification of Incident Cases of Infection [J]. Clin Infect Dis, 2021, 73(5): 842-849.10.1093/cid/ciab140.]. That meant patients with high viral load had high transmission rate, but the rate would de-crease after they were identified, because detected patients with HIV initiated ART immediately, which generally led to undetectable virus. Thus, the majority of participants in this study appeared scattered.

  • “We did not identify significant differences between the recent infection group and long-term infection group at baseline, indicating a small sample size in this study.” (Lines 266-267). What differences were evaluated? This was not mentioned in the results or methods.

We evaluated socio-demographics (including age, ethnicity, marriage, education, occupation, subtype, and whether cluster or not) and gene subtype.

Minor comments:

Materials and Methods:

  • A justification of the 1.5% cutoff should be included, not just a reference to earlier uses.

 Thanks for your kind advice. We added an analysis of 0.5% cutoff in the revised manuscript.

Discussion:

  • HARRT should be HAART (line 225), and its true reference stated (highly active antiretroviral therapy. Otherwise, just use ART abbreviation.

 This abbreviation was revised in the manuscript.

Comments on the Quality of English Language

As stated above, grammar issues throughout this manuscript (particularly in the discussion) make it very difficult to read.

Thanks for your kind advice. We performed additional editing to correct the grammar issues, particularly in the discussion.

Reviewer 3 Report

Zhou et al. provided an interesting piece of the puzzle on HIV epidemiology in China. Before publication is considered, I have a few recommendations on how the manuscript could be further improved.

Major comments.

1. General comment. The authors conclude that the detection of HIV transmission clusters calls for public health enforcement to break the transmission cycle. As the authors are most certainly aware of, there is no global agreement that – mostly prohibitive – public health measures are the most efficient response to the HIV pandemic. Other procedures, predominantly early antiretroviral therapy rapidly pushing the viral load below the detection threshold combined with ready access to pre-exposure prophylaxis for individuals at risk of acquiring HIV, are efficient as well and associated with good compliance due to their non-prohibitive nature which does not interfere with the affected individuals’ privacy. The authors may want to at least consider this alternative option, making their conclusions more globally acceptable.

2. Introduction, first paragraph. The dominance of men-having-sex-with-men in HIV epidemiology is not globally equally distributed. There are regions in the world, in particular in Sub-Saharan Africa, there infected women quantitatively dominate, stressing a regional dominance of the heterosexual transmission route. The authors may want to weaken their respective statement accordingly.

3. Materials and methods. The applied statistical assessments should be provided in more detail under the “Statistical analysis” sub-heading. It is insufficient just to mention that the R software package has been used.

4. Discussion. The authors should discuss the assumptions used to define the genomic similarity thresholds, based on which the transmission networks were established.

Minor comments.

5. Please make sure that abbreviations are spelled out at first use.

6. Although I am not an English native speaker myself, some of the phrases and Grammar sounded non-idiomatic to me. I recommend to have the manuscript checked by a native speaker – either from the side of the authors or from the side of the journal – before final publication is considered.

Author Response

Major comments.

  1. General comment. The authors conclude that the detection of HIV transmission clusters calls for public health enforcement to break the transmission cycle. As the authors are most certainly aware of, there is no global agreement that – mostly prohibitive – public health measures are the most efficient response to the HIV pandemic. Other procedures, predominantly early antiretroviral therapy rapidly pushing the viral load below the detection threshold combined with ready access to pre-exposure prophylaxis for individuals at risk of acquiring HIV, are efficient as well and associated with good compliance due to their non-prohibitive nature which does not interfere with the affected individuals’ privacy. The authors may want to at least consider this alternative option, making their conclusions more globally acceptable.

The detection of HIV transmission clusters may well be effective for identifying larger unexpected outbreaks, identifying key populations for prevention resources and monitoring the effect of prevention interventions, than for prospectively identifying and interrupting incident transmission events in epidemics.

  1. Introduction, first paragraph. The dominance of men-having-sex-with-men in HIV epidemiology is not globally equally distributed. There are regions in the world, in particular in Sub-Saharan Africa, there infected women quantitatively dominate, stressing a regional dominance of the heterosexual transmission route. The authors may want to weaken their respective statement accordingly.

Our study population is MSM, one of key population at high risk of HIV. We hope our study is helpful to control HIV transmission among MSM.

  1. Materials and methods. The applied statistical assessments should be provided in more detail under the “Statistical analysis” sub-heading. It is insufficient just to mention that the R software package has been used.

Main R packages used included “dplyr”, “broom”, “mapdata”, “maps”, “maptools”, “rgdal”, “ggplot2”, “plyr”, “cowplot”, “tidyverse”, "ape" and “sf”.

  1. Discussion. The authors should discuss the assumptions used to define the genomic similarity thresholds, based on which the transmission networks were established.

We performed analysis for different distance thresholds (ranging from 0.1% to 3.25%) to determine the optimal threshold (i.e. the distance when the maximum ratio of the number of clusters to distance is detected). The ratio at 1.2% distance threshold was greater than 1.5%, and the number of clusters at both distances were the same. Then, we chose a 1.5% distance threshold based on previous studies. Since the aim of this study is relative to RHI, we chose the genomic distance thresholds of 0.5% and 1.5%.

Minor comments.

  1. Please make sure that abbreviations are spelled out at first use.

We have revised these mistakes in the manuscript.

  1. Although I am not an English native speaker myself, some of the phrases and Grammar sounded non-idiomatic to me. I recommend to have the manuscript checked by a native speaker – either from the side of the authors or from the side of the journal – before final publication is considered.

Thanks for your kind advice.